# The Impact of Neurophysiological Monitoring during Intradural Spinal Tumor Surgery

**DOI:** 10.3390/cancers16122192

**Published:** 2024-06-11

**Authors:** Furkan Ilhan, Sébastien Boulogne, Alexis Morgado, Corentin Dauleac, Nathalie André-Obadia, Julien Jung

**Affiliations:** 1Neurophysiology & Epilepsy Unit, Neurological Hospital P. Wertheimer, Hospices Civils de Lyon, 59 Boulevard Pinel, 69677 Bron, France; furkan.ilhan@chu-lyon.fr (F.I.); sebastien.boulogne@chu-lyon.fr (S.B.); nathalie.obadia-andre@chu-lyon.fr (N.A.-O.); 2Tiger TEAM, INSERM U1028, UMR5292, Lyon Neuroscience Research Center, CNRS, University Claude Bernard Lyon 1, 69675 Lyon, France; 3Neurosurgical Department, Neurological Hospital P. Wertheimer, Hospices Civils de Lyon, 59 Boulevard Pinel, 69677 Bron, France; alexis.morgado@chu-lyon.fr (A.M.); corentin.dauleac@chu-lyon.fr (C.D.); 4NeuroPain Lab, INSERM U1028, UMR5292, Lyon Neuroscience Research Center, CNRS, University Claude Bernard Lyon 1, 69675 Lyon, France; 5EDUWELL Team, INSERM U1028, UMR5292, Lyon Neuroscience Research Center, CNRS, University Claude Bernard Lyon 1, 69675 Lyon, France

**Keywords:** spinal cord tumor, intraoperative neuromonitoring, evoked potentials, prognosis, ependymoma, meningioma, Schwannoma

## Abstract

**Simple Summary:**

Spinal cord tumors represent a significant surgical challenge. Neurophysiological evaluation using motor evoked potentials (MEPs) and somatosensory evoked potentials (SSEPs) is used to assess the function of motor and somatosensory pathways before and during surgery. However, its role with respect to the detection of neurological damage is not fully known. This study assesses the prognostic value of alterations of pre- and per-operative MEPs and SSEPs in a cohort of 67 patients undergoing spinal cord tumor surgery. We show that pre- and intraoperative alterations to MEPs and SSEPs are strongly associated with a risk of neurological deterioration 3 months after surgery that is independent of initial clinical severity, tumor size, or histological subtype. Leveraging a classical machine learning approach, it is possible to predict neurological outcomes with an accuracy of 84%. Thus, this study demonstrates the pivotal role of pre- and intraoperative neurophysiological explorations in detecting and preventing neurological suffering during spinal tumor surgery.

**Abstract:**

Surgery for spinal cord tumors poses a significant challenge due to the inherent risk of neurological deterioration. Despite being performed at numerous centers, there is an ongoing debate regarding the efficacy of pre- and intraoperative neurophysiological investigations in detecting and preventing neurological lesions. This study begins by providing a comprehensive review of the neurophysiological techniques commonly employed in this context. Subsequently, we present findings from a cohort of 67 patients who underwent surgery for intradural tumors. These patients underwent preoperative and intraoperative multimodal somatosensory evoked potentials (SSEPs) and motor evoked potentials (MEPs), with clinical evaluation conducted three months postoperatively. The study aimed to evaluate the neurophysiological, clinical, and radiological factors associated with neurological outcomes. In univariate analysis, preoperative and intraoperative potential alterations, tumor size, and ependymoma-type histology were linked to the risk of worsening neurological condition. In multivariate analysis, only preoperative and intraoperative neurophysiological abnormalities remained significantly associated with such neurological deterioration. Interestingly, transient alterations in intraoperative MEPs and SSEPs did not pose a risk of neurological deterioration. The machine learning model we utilized demonstrated the possibility of predicting clinical outcome, achieving 84% accuracy.

## 1. Introduction

### 1.1. Context

Primary spinal cord tumors (SCT) are rare, accounting for only 5–12% of central nervous system tumors [1]. Extradural tumors typically stem from metastatic origins, while intradural tumors encompass a spectrum of lesions including intramedullary (20–30%) and extramedullary (70–80%) entities, ranging from benign meningiomas to more aggressive intramedullary gliomas [2]. The majority of SCTs manifest as benign and slow-evolving neoplasms. Consequently, the established treatment involves total removal [3]. While this may yield long-term survival, it necessitates a delicate balance with the risk of permanent postoperative neurological deficits. Early studies revealed notable postoperative deficits, with dorsal column dysfunction rates reaching up to 65% [4] and quadriplegia occurring in up to 3.7% of cases [5]. Recent research indicates that up to 11% of patients experience significant permanent clinical worsening [6]. The intricate pathways of the spinal cord play a pivotal role in sensory and motor signal transmission. SCTs have the potential to disrupt these pathways, rendering the challenge of surgical removal multifaceted, requiring not only tumor excision but also preservation of spinal cord integrity to prevent neurological damage. Additionally, tumor-induced anatomical distortions can complicate midline identification for myelotomy, necessitating a meticulous and nuanced approach by the surgical team.

In this context, neurophysiology has emerged as a transformative adjunct for the clinical management of spinal cord tumors and intraoperative decision-making. Preoperative evoked potentials can assess the extent of spinal cord dysfunction, while somatosensory evoked potentials (SSEPs) and motor evoked potentials (MEPs) evaluate dorsal and lateral column function, respectively [7]. They can complement clinical examinations in cases with few symptoms, and may impact surgical decision-making. When altered, they aid in identifying patients at higher risk of neurological deficits, potentially influencing preoperative planning [8,9]. Additionally, they provide the surgeon with insight into the feasibility of intraoperative neurophysiologic monitoring (IONM), thereby informing decision-making regarding its potential contribution to surgical safety.

IONM facilitates real-time assessment of neural function during surgery, enabling the detection of impending neural compromise. This can prompt the surgical team to reevaluate their approach or implement corrective measures to mitigate the risk of irreversible spinal cord lesions [10]. Such measures may include spinal cord warming, correction of arterial hypotension, or temporarily suspending the procedure. This dynamic feedback enhances the safety of the resection while offering valuable insights into the intricate interplay between surgical maneuvers and neural structures.

IONM protocols can be customized to each patient and the specific characteristics of the SCT, ensuring a personalized surgical approach [11]. Successful implementation of IONM depends on effective collaboration among the surgeon, anesthesiologist, and neurophysiologist.

### 1.2. IONM Techniques

A schematic representation of neurophysiological techniques is shown in Figure 1.

#### 1.2.1. Anesthesia Requirements

Anesthetic agents can impact the recorded potentials, necessitating careful management to ensure accurate monitoring [11]. Management should involve a continuous infusion of propofol and fentanyl, with minimal use of bolus infusion. Paralytics and halogenated anesthetics should be avoided, as they may inhibit the recording of MEPs.

#### 1.2.2. Somatosensory Evoked Potentials (SSEPs)

SSEPs were the pioneering IONM technique utilized for SCT [12]. They evaluate the integrity of the dorsal columns. Electrical stimulation is applied to a peripheral nerve, typically the posterior tibial nerve for lower extremity monitoring and the median nerve for upper extremity monitoring. This elicits sensory impulses that travel along the ascending sensory pathways through the peripheral nerves and the spinal cord, with cortical responses recorded on the scalp. Changes in SSEPs during surgery may indicate compromise to the sensory pathways. Common alert criteria signaling potential spinal cord injury include a decrease in amplitude of over 50% or an increase in latency of more than 10% [13].

#### 1.2.3. Motor Evoked Potentials (MEPs)

MEPs began to see application in the mid-1990s [14] for evaluating the integrity of the motor pathway, primarily originating from the motor cortex and traversing through the spinal cord to innervate muscles. Transcranial electrical stimulation is administered to the motor cortex, inducing motor impulses that propagate along the descending motor pathway. Responses are captured via electromyographic (EMG) electrodes positioned in target muscles, enabling the monitoring of muscle responses to motor stimuli. The selection and quantity of target muscles are customized based on the location of the SCT and preoperative assessment. Alert criteria may vary among studies; some employ an all-or-nothing approach [15], while others consider an amplitude decrease exceeding 50% [13,16].

Figure 2 shows an example of an IONM alert.

#### 1.2.4. D-Waves (DW)

D-waves, also known as direct motor responses, represent another facet of MEP monitoring, directly assessing the functional integrity of the lateral columns in the spinal cord. Electrical stimulation can either be transcranial on the motor cortex or applied directly to the spinal cord cranial to the operating site, bypassing the motor cortex. The response is captured on an electrode positioned caudal to the operating site in the epidural space after laminotomy. Consequently, D-waves cannot be recorded below the level of T12 due to the lack of sufficient spinal cord material for recording. While D-wave recordings are robust they are unfortunately not always achievable. Nevertheless, they are regarded as the most reliable predictor of long-term motor outcomes; stability in D-wave recordings indicates unchanged motor status if MEPs have been preserved or transient mono/paraparesis if MEPs have been altered. Conversely, while loss of D-waves is strongly associated with postoperative motor deficits, it does not pinpoint which side of the lateral columns is impaired [6,16,17]. Nevertheless, D-waves should consistently accompany MEP recordings. MEPs hold the advantage of being more reliably recordable than D-waves throughout the surgery and more sensitive to unilateral motor damage, while also offering more precise topographical information [10].

#### 1.2.5. Multimodal Monitoring

Signal processing techniques are utilized to filter and analyze the recorded neurophysiological signals. A stable baseline is established prior to surgical resection, serving as a reference for comparison throughout the procedure. The utilization of multimodal IONM incorporating SSEPs, MEPs, and D-waves recordings enhances accuracy in detecting spinal cord impairment, as surgery may selectively affect either the somatosensory or motor pathways [8,10,18]. D-waves and SSEPs can be continuously monitored throughout surgery, whereas MEPs are recorded on-demand due to their potential to induce muscle twitching, which may disrupt the surgical process. IONM alerts raise greater suspicion of spinal cord impairment when multiple IONM modalities are engaged and when changes are sudden and prolonged, surpassing the typical spontaneous curve-to-curve variability [10].

### 1.3. Objectives of the Study

Numerous retrospective surgical series on SCT surgery have been published. Due to the rarity of this disease, most series are small and do not consider preoperative neurophysiological assessment. The utility of IOM remains a subject of debate [19,20]. Our objective is to augment the existing literature by investigating a cohort of 67 consecutive SCT patients who underwent surgery over a seven-year period at the University Hospital of Lyon spanning from 2016 to 2023. The study encompasses preoperative clinical, MRI, and neurophysiological assessments along with IONM techniques and alerts, which are juxtaposed with postoperative clinical and radiological evaluations. The aim is to analyze the pre- and intraoperative predictors of optimal outcomes.

## 2. Materials and Methods

The study was conducted in accordance with the Declaration of Helsinki, and the protocol was approved by the Internal Ethics Review Board 00013204 of Hospices Civils de Lyon (protocol code 23-365, date of approval 21 December 2023). Informed consent was obtained from all patients involved in the study. All data were extracted from medical records.

### 2.1. Patients

Patients admitted to our university hospital for surgical treatment of a spinal cord lesion and who had undergone preoperative MEPs and SEPs, intraoperative neurophysiological monitoring (IONM), and clinical evaluation at 3 months between April 2016 and May 2023 were consecutively selected and included. Exclusion criteria comprised age under 18 years old, having solely undergone spine surgery without intracanal procedures, and having a lesion below the cauda equina level.

### 2.2. Electrophysiological Recordings

#### 2.2.1. Preoperative MEPs and SSEPs

The preoperative functional integrity of the corticospinal pathway was evaluated through MEPs, employing transcranial magnetic stimulation (TMS) in accordance with the IFCN guidelines [21]. A magnetic stimulator (Mag-Pro X100, MagVenture©, Farum, Denmark) administered transcranial and motor root stimulation. The latencies of the responses to these two stimulation modalities corresponded to the total motor conduction time (TMCT) and peripheral magnetic motor conduction time (PMCTm), respectively. Peripheral electrical stimulation was utilized to record motor distal peripheral responses (M-waves) and F-waves, enabling calculation of the electrical peripheral conduction time (PMCTe) [21]. Preoperative MEPs were recorded, at minimum, on the abductor digiti minimi and tibialis anterior for the upper and lower limbs, respectively. Central motor conduction time was computed using two methodologies: TMCT–PMCTm and TMCT–PMCTe. Additionally, the amplitude ratio between responses to cortical magnetic and M-waves was determined. The primary pathological criteria included abolished responses to transcranial magnetic stimulation with preserved peripheral responses or prolonged conduction time.

The preoperative functional integrity of the lemniscal sensory pathway was evaluated via SSEPs. These were recorded using equipment from Micromed (Mâcon, France) employing peripheral electrical stimulation (median and tibial nerve) and staged recording by Ag–AgCl surface electrodes to assess conduction from the peripheral nerve through the dorsal column to the sensory cortex [22]. The primary pathological criteria included the absence of cortical responses (N20 for upper limbs and P39 for lower limbs) or prolongation of spinal conduction time. Spinal conduction time was estimated by the difference between the latencies of plexual and cervico-bulbar responses (P9–P14 for upper limbs or N22-P30 for lower limbs) when cervico-bulbar responses were detectable. In cases where cervico-bulbar responses were absent or highly desynchronized, spinal conduction time was estimated by the difference between the latencies of the plexual and parietal responses (P9-N20 for upper limbs or N22-P39 for lower limbs).

Preoperative MEPs and SSEPs were recorded 2 days before surgery.

#### 2.2.2. Intraoperative Neurophysiological Monitoring

For details on the monitoring methodology, we refer readers to Legatt [23]. Anesthesia was performed with propofol (6–8 mg/kg/h) and remifentanil (0.15–2 μg/kg/min) during the entire surgical procedure, without curarization beyond the induction phase. All patients were maintained as normothermic and normotensive. For all patients, IONM was performed using the NIM-ECLIPSE® NS System (Medtronics, Minneapolis, MN, USA).

Intraoperative SSEPs were elicited through stimulation of the tibial and/or median nerves at motor threshold. Somatosensory cortex responses, namely, N20 and P39, were recorded at P3/P4 or C’z for the upper and lower limbs, respectively. Baseline curves, comprising an average of at least 200 responses, were consistently established before incision. Throughout the surgery, N20 (for median SSEP) and P39 (for tibial SSEP) amplitudes and latencies were continuously monitored. Alert criteria were defined as an amplitude reduction exceeding 50% compared to the baseline and/or an N20 or P39 latency prolongation over the baseline by more than 10% (Sala et al., 2006) [13].

Intraoperative MEPs were elicited by a train of transcranial electrical stimulations delivered with corkscrew electrodes at C1 and C2, according to the international 10–20 EEG system. MEPs were recorded from the biceps brachii and abductor digiti minimi muscles in the upper limbs and from the vastus lateralis, tibialis anterior, and extensor digitorum brevis muscles in the lower limbs. The peak-to-peak amplitude of the MEPs was monitored during the entire surgery. The alert criteria were defined as an amplitude reduction of more than 50% compared to the baseline in at least two muscles of the same limb [13,16].

When feasible, the surgeon positioned an epidural two-contact electrode just caudal to the surgical site following laminectomy to capture the direct descending volleys of the pyramidal tracts (D-Wave DW). This response was induced with a single transcranial electrical stimulation and continuously monitored. A decrease of 50% or more in the baseline amplitude was deemed significant. MEPs and DW responses were combined to determine motor alerts. A noteworthy alert in either MEPs, DW responses, or both was considered to constitute a significant MEPs–DW motor alert.

In the event of significant alterations in the evoked potentials, the absence of technical errors or changes in the anesthetic parameters was initially verified. Subsequently, neurosurgeons were promptly notified to ensure that the surgical approach could be adjusted accordingly. This could involve temporarily shifting surgical manipulation to a different area, irrigating with warm saline solution, or addressing hypotension. If there was bilateral loss of MEPs and/or insufficient recovery of D-wave amplitude by over 50%, then the surgery was halted; however, isolated decrements in SSEPs were not grounds to abandon surgery, as SSEPs are highly sensitive to surgical manipulation even prior to tumor removal through the approach between the dorsal columns.

### 2.3. Data Extraction

#### 2.3.1. Clinical Characteristics

For each patient, the study collected the following characteristics: age at the time of surgery, gender, preoperative functional score using the modified McCormick scale (MMS), Karnofsky index, postoperative functional score three months after surgery using MMS, and presence or absence of functional deterioration in each limb both immediately after surgery and three months post-surgery. Preoperative clinical status was categorized based on the MMS as mild (grades 1 and 2) or severe (grades 3, 4, and 5), and postoperative clinical status was similarly defined.

The primary outcome of the study was the overall clinical outcome of the patient as assessed three months after surgery. Deterioration was defined as an increase of at least one point in MMS or significant somesthetic functional deterioration (complete hypoesthesia and/or ataxia), while clinical stability or improvement indicated stable condition.

The secondary outcome focused on the functional outcome of each limb three months post-surgery. Functional deterioration was defined as motor (worsening of motor deficit) and/or somesthetic (sensory deficit or ataxia) deterioration compared to the preoperative state, while clinical stability or improvement indicated stable condition.

#### 2.3.2. Anatomopathological Characteristics

Anatomopathological data were grouped into three main groups: meningioma, ependymoma, Schwannoma, and one group with rarer and more heterogeneous lesions (including arachnoid cyst, lipoma, cavernoma, and astrocytoma).

#### 2.3.3. Radiological Characteristics

Based on preoperative MRIs, the following elements were collected: tumor location (intradural extramedullary, intradural intramedullary), tumor level (cervical, thoracic, lumbar, or a combination of these levels), rostro-caudal size of the solid part of the lesion, and presence or absence of a syringomyelic cavity. Lesions were grouped into high (cervical or cervico-thoracic) or low (thoracic or thoraco-lumbar) levels. The choice of these two tumor levels was motivated by the need to perform IONM adapted to these two groups (see below). Postoperative MRIs performed at 3 months were used to identify the presence or absence of residual tumoral lesions.

#### 2.3.4. Intraoperative Evoked Potentials

First, all instances of anomalies meeting the alert criteria defined above were compiled for each limb for both MEPs–DW and SSEPs. Thus, alerts occurring at the individual limb level were categorized into three groups: (i) any SSEPs alert, (ii) any MEPs–DW alert, and (iii) any MEPs–DW/SSEPs alert.

Second, several groups of alerts were defined at the patient level: (i) any MEPs–DW alert, any SSEPs alert, and any MEPs–DW/SSEPs alert (comprising all alert instances, whether transient or persistent, affecting at least one limb for the considered modality), (ii) transient alerts (encompassing all instances of alerts for at least one limb that resolved during surgery for the considered modality), and (iii) persistent alerts (encompassing all instances of alerts for at least one limb that persisted until the end of surgery for the considered modality).

### 2.4. Statistical Analysis

#### 2.4.1. Primary Outcome

The primary outcome focused on the presence of clinical deterioration assessed three months after surgery.

First, the association between the primary outcome and intraoperative evoked potentials was examined. The predictive capacity of each group of intraoperative electrophysiological anomalies to anticipate clinical outcomes at three months was assessed using conventional diagnostic performance measures, including sensitivity, specificity, positive predictive value, negative predictive value, and accuracy.

Second, univariate analyses were performed to investigate the relationship between clinical, radiological, anatomopathological, and electrophysiological variables and clinical outcomes at three months. This involved utilizing the chi-squared test, Mann–Whitney U test, or Kruskal–Wallis test, as appropriate.

Third, a multivariate logistic regression model was employed to identify variables associated with clinical outcomes at three months. Only variables that were significant at *p* < 0.1 in univariate analyses were included in the final model, with the significance level set at *p* < 0.05.

Finally, a classical machine learning approach was employed using a k-nearest neighbors (KNN) classifier to predict clinical outcomes at 3 months [24]. The selection of relevant predictive variables was based on univariate analyses, retaining only variables significant with a *p*-value < 0.1. The dataset was divided into training (70% of data) and test sets (30%) for model fitting and evaluation, respectively. Five-fold cross-validation was performed to ensure the generalization of results and validate the stability and reliability of the model’s performance. The entire approach was implemented using the scikit-learn module (https://scikit-learn.org/stable/, accessed on 3 March 2024).

#### 2.4.2. Secondary Outcome

The potential association between the occurrence of motor or somesthetic neurological degradation specific to a limb and the presence of an IONM alert for that limb was explored. This investigation aimed to assess the topographical specificity of IONM alterations in predicting functional outcomes at three months. Classical diagnostic performance measures (including sensitivity, specificity, positive predictive value, negative predictive value, and accuracy) were calculated for each group of alerts (any SSEPs alert, any MEPs–DW alert, and any MEPs–DW/SSEPs alert) to predict individual limb deficits at three months.

Subsequently, univariate analyses were conducted to examine the relationship between clinical, radiological, anatomopathological, and electrophysiological variables and clinical outcomes at three months, using the chi-squared test and Mann–Whitney U test where applicable.

Finally, a multivariate logistic regression model was utilized to identify factors predictive of individual limb functional outcomes. Only factors significant at *p* < 0.1 in univariate analyses were included in the final model.

## 3. Results

### 3.1. Patient Characteristics

During the study period, 67 patients (43 men and 24 women) who had undergone spinal cord surgery, preoperative, and IONM were included (Table 1). The mean age of the patients was 50 years (±14). Preoperatively, 44 patients (64%) had a mild neurological deficit and 24 patients (36%) had a severe deficit. The mean rostro-caudal size of the solid portion of the tumor was 32 mm (±20). For 32 patients (48%) the lesion was at a high level, while for 35 patients the lesion was at a low level (52%). In 40 patients (60%), the lesion was intradural extramedullary, while for 27 patients (40%) it was intradural intramedullary.

Regarding histology, 24 patients (35%) had an ependymoma, 17 patients (25%) had a meningioma, 14 patients (21%) had a Schwannoma, and 12 patients had other etiologies (two hemangioblastomas, two neurofibromas, two cavernomas, two lipomas, two pilocytic astrocytomas, one neuroenteric cyst, and one metastasis of a lung carcinoid tumor). Postoperative MRI showed complete tumor resection in 52 patients (79%), while 14 patients (21%) had a residual lesion. Three months after resection, neurological examinations were stable or improved for 45 patients (67%), while 22 patients (33%) showed deterioration. Among those patients with clinical deterioration, three patients worsened from MMS 1 to 2, eleven patients from MMS 2 to 3, four patients from MMS 3 to 4, and one patient from MMS 4 to 5, while three patients had the same MMS but with increased somatosensory deficit.

### 3.2. Overall Results of Electrophysiological Investigations

In 25 patients (37%), preoperative evoked potentials were impaired, while in 42 patients (63%) they were normal (Table 1). In the preoperative period, there was a significant clinico-electrophysiological association between the preoperative clinical status and any neurophysiological alterations of either MEPs or SSEPs (*p* = 0.001, Chi2 test).

In terms of IONM, MEPs were obtained and monitored during surgery in all 67 patients. The number of muscles monitored per patient varied from 5 (min) to 10 (max). DW were obtained and monitored during surgery in 24 patients. SSEPs were not recordable in seven patients; thus, as a whole, IONM was possible either with MEPs or SSEPs or combined MEPs and SSEPs in all patients.

Any alerts (transient or persistent) occurred in 57% of the patients, with persistent alerts occuring in 42% and transient alerts only in 15%. Regarding MEPs-DW, 55% showed no modification, while 45% presented an alert (36% of persistent alerts and 9% of transient alerts). Regarding SSEPs, 63% showed no modification, while 36% presented an alert (23% of persistent alerts and 13% of transient alerts).

A significant association was found between alteration of preoperative evoked potentials and persistent IONM alterations (*p* = 0.002, Chi2 test), while no association was found between the occurrence of a transient IONM alert and preoperative evoked potentials (*p* = 0.11, Chi2 test).

### 3.3. Relationship between Patient’s Clinical Outcome Three Months after Surgery and Neurophysiological Investigations

To assess the primary outcome, as a first step, the relationship between the overall clinical status of patients 3 months after surgery and the diagnostic performance of IONM was evaluated. Table 2 displays the sensitivities, specificities, positive and negative predictive values, and accuracy of IONM alterations in predicting clinical deterioration at 3 months.

Overall, the sensitivity of alterations in different IONM modalities (MEPs-DW, SSEPs, or alterations in either MEPs-DW or SSEPs) ranged from 0 to 95%, the specificity from 72 to 92%, the positive predictive value from 0 to 79%, and the negative predictive value from 78 to 97%. The positive predictive value of MEPs-DW was generally higher than that of SSEPs and combined MEPs-DW/SSEPs monitoring. However, the highest negative predictive values were achieved with the combined use of MEPs-DW/SSEPs (96% for any alterations and 97% for persistent alerts). Persistent IONM alterations were associated with higher positive predictive value compared to any alterations, though with a comparable negative predictive value. The best accuracy was achieved with persistent MEPs-DW alteration (0.88) and with persistent alteration in combined MEPs-DW/SSEPs monitoring (0.88). Regarding transient IONM alerts, which occurred in only ten patients, none of them were associated with clinical deterioration (meaning that the positive predictive value was 0); however, the negative predictive value was high (between 78 and 96%).

Moreover, no association was found between the occurrence of any IONM alterations or persistent IOM alterations and the presence of residual MRI lesions (*p* > 0.05, Chi2 test).

In addition, univariate analyses were conducted to explore variables associated with the clinical outcome at 3 months (Table 3). Several factors were found to be significantly associated with clinical deterioration at 3 months: the rostro-caudal size of the solid portion of the lesion on preoperative MRI (*p* = 0.01), the presence of abnormalities in preoperative evoked potentials (*p* = 0.03), the occurrence of any alert during IONM (*p* < 0.001) or persistent abnormalities during IONM (*p* < 0.001), the histological type of the lesion (*p* = 0.001) with an increased risk of deterioration for ependymomas, and the presence of residual lesion on postoperative MRI (*p* < 0.001). Other factors were not related to the clinical status at 3 months: patient age or gender, high or low location of the lesion, intramedullary or extramedullary location of the lesion, and preoperative functional clinical status (*p* > 0.05 for each of these factors). In multivariate analysis using logistic regression, only the presence of abnormalities in preoperative evoked potentials (*p* = 0.04) or an alert (*p* < 0.001) or persistent abnormalities during IONM (*p* < 0.001) were associated with clinical deterioration at 3 months (Table 3).

Finally, a classification model for the clinical outcome at 3 months was implemented using a KNN model, with the following variables as predictors: lesion size, lesion level, presence of abnormalities in preoperative evoked potentials, occurrence of an alert during IONM, and lesion histology. The model’s performance was evaluated after validation through 5-fold cross-validation. The KNN model demonstrated an accuracy of 0.80 and roc-AUC of 0.84, emphasizing its effectiveness in data classification.

### 3.4. Topographical Specificity of IONM Alterations

Overall, the diagnostic performance of IONM for prognostication of clinical degradation of individual limbs at 3 months following surgery was rather high (Table 4). The sensitivity of alterations in different IONM modalities (MEPs, SSEPs, or alterations in either MEPs or SSEPs) ranged from 42 to 95%, the specificity from 72 to 83%, the positive predictive value from 42 to 58%, and the negative predictive value from 83 to 92%. The link between the functional evolution of each monitored at-risk limb during surgery and IONM alterations was evaluated through univariate and multivariate analyses (Table 5). A total of 200 limbs were analyzed, including 152 with a stable clinical status and 48 showing deterioration at 3 months. In univariate analysis, lesion size (*p* = 0.001), lesion level (*p* = 0.01), occurrence of an alert during IONM for that limb (*p* < 0.001), and lesion histology (*p* < 0.001) with a higher risk for ependymomas were all associated with the occurrence of clinical deterioration (either motor or sensory) at 3 months for the considered limb. In multivariate analysis, only the occurrence of an alert during IONM for that limb (*p* < 0.001) remained associated with the occurrence of clinical deterioration.

## 4. Discussion

The intricate relationship between these tumors and the lengthy spinal cord tracts exposes patients to the risk of mechanical or ischemic damage during surgery. Given the rarity of these tumors, a handful of neurosurgeons encounter only a small number of cases annually, resulting in an extended learning curve. Consequently, surgery involving spinal cord tumors remains challenging even for seasoned surgeons. Key obstacles include determining the optimal timing for surgery and achieving maximal resection while minimizing the likelihood of postoperative neurological deficits. In this regard, intraoperative neurophysiological monitoring (IONM) emerges as an invaluable technique, offering real-time functional insights into the patient’s spinal cord health throughout the surgical procedure [10].

In our single-center cohort study, a total of 67 patients underwent a standardized protocol, which included clinical evaluation, MRI imaging, preoperative evoked potentials, IONM, and surgical intervention. Preoperatively, approximately two-thirds of the patients exhibited moderate functional disability (MMS 1 or 2). In terms of tumor characteristics, around 60% of the patients had intradural extramedullary tumors, while 40% had intradural intramedullary tumors. A significant portion of our patient population can be classified as at high risk for deficit, as noted in previous studies [25]. Specifically, 22 patients experienced clinically significant neurological deterioration three months post-surgery, manifesting as either worsening somatosensory or motor function. It is important to acknowledge that our criteria for detecting neurological deterioration were highly sensitive; any increase in MMS score or somatosensory deficit was considered as clinical deterioration. Using more stringent criteria, such as severe deterioration impacting walking ability, would likely result in a lower rate of neurological deterioration. Additionally, the evaluation of clinical status was conducted three months postoperatively, and it is worth noting that functional recovery can continue beyond this timeframe. Regarding histological data, the most common etiologies observed were ependymomas, meningiomas, and Schwannomas, which align with reported frequencies for these types of lesions.

The most notable finding of our study underscores the significance of IONM in detecting intraoperative lesions that may lead to chronic neurological deficits. Indeed, the presence of any alert during monitoring was correlated with a deterioration in neurological status at three months in 54% of patients with SSEPs, 63% of those with MEPs-DW, and 55% of those undergoing combined SSEPs-MEPs-DW monitoring. Utilizing such combined monitoring enhances the likelihood of identifying neurological lesions, thereby boosting the sensitivity of combined neuromonitoring, consistent with findings from several previous studies [11,26,27,28]. When an IONM alert persists until the end of surgery, a deterioration in neurological status is observed much more frequently, occurring in 78% of cases with SSEPs, 79% with MEPs-DW, and 75% of patients undergoing combined SSEPs-MEPs-DW monitoring. Furthermore, our study underscores the topographical specificity of IONM alterations; when IONM alterations affect a specific limb during surgery, there is a heightened risk of immediate postoperative deficit in the same limb. This novel finding has not, to the best of our knowledge, been reported previously, and strongly supports a mechanistic link between the dynamic evolution of IONM and the functional status of spinal tracts. From a surgical standpoint, IONM alerts can serve to guide the neurosurgeon regarding anatomical sites that may have induced alterations, thereby allowing for adaptation of the surgical procedure accordingly. Additionally, the negative predictive value of various IONM modalities is notably high, ranging between 80% and 95%, particularly for the SSEPs-MEPs-DW combination. This suggests that the absence of alterations during IONM provides a reliable indication of the absence of neurological lesions and should offer valuable feedback to the surgeon, enabling the pursuit of the most optimal tumor resection. These results align with findings from several robust Class I–II studies demonstrating that IONM is a valid diagnostic tool for predicting spinal cord injury [19,20,29]. A recent meta-analysis conducted by Rijs et al. using data collected from 31 studies focusing on intramedullary tumors showed diagnostic performances of IONM in the same range as the present study (sensitivity ranging between 0.81–0.83 and specificity between 0.59–0.83, depending on the use of MEPs-DW, SSEPs, or multimodal monitoring) [18]. Before the widespread adoption of IONM, the primary method for assessing spinal cord integrity during surgery was the wake-up test [30]. However, in cases of spinal cord tumor surgery, lesions affecting motor tracts are often localized or may only involve a single limb, leading to potential false negative results with the wake-up test. Nowadays, the wake-up test is increasingly being replaced by IONM, which offers greater sensitivity in detecting minor spinal cord lesions. The wake-up test may still be employed when IONM results are unreliable or when there are lingering questions about the patient’s motor status even after IONM results show changes.

Despite the high predictive value of IONM changes, the prediction of clinical deterioration is not flawless, even for permanent changes. It is important to note that in this study we only considered neurological deficits at three months postoperatively, and it is conceivable that the predictive value immediately after surgery would be higher. Therefore, IONM changes during spinal cord tumor surgery should be regarded as a significant indicator of the risk of clinical deterioration, particularly for permanent changes. However, their interpretation should be integrated with all clinical, radiological, and surgical factors associated with the risk of neural lesion. Indeed, while univariate analysis highlights several factors linked to an increased risk of postoperative deficit (such as larger lesion rostro-caudal size, presence of a cervical tumoral component, ependymoma histology, presence of postoperative MRI residual lesion, and alteration of preoperative evoked potentials), preoperative neurophysiological assessment and IONM alterations emerge as the only significant factors remaining after multivariate analysis. While it is already known that lesion size, preoperative clinical status, and intramedullary localization of the tumor are associated with the risk of deficit, the prognostic value of preoperative evoked potentials is not well documented [31,32]. Two recent studies have shown that alterations in preoperative evoked potentials are associated with a higher risk of IONM alteration [8,33]; however, to the best of our knowledge, the close relationship between preoperative spinal cord functional assessment and long-term clinical status has not been specifically evaluated. In this regard, our findings suggest that evoked potentials should be incorporated into the preoperative assessment of asymptomatic or pauci-symptomatic patients for several reasons: (i) Preoperative evoked potentials can serve as a tool to detect neurological dysfunction before surgery, enabling early intervention in slowly evolving tumors before functional impairment exacerbates and jeopardizes post-operative recovery chances; (ii) alterations in preoperative evoked potentials indicate a heightened risk for surgery, providing valuable insight into the potential complications and informing decision-making regarding the surgical approach; and (iii) preoperative evoked potentials are instrumental in assessing the feasibility of IONM, offering valuable information that can guide the implementation and optimization of intraoperative monitoring techniques. Thus, we utilized a machine learning model approach to classify clinical outcomes at three months using multivariate data. While machine learning models are increasingly applied across various clinical domains, their current use in the context of spinal tumors remains relatively nascent [34]. Only two published studies have focused on clinical outcome of spinal tumors, and IONM was not included among their predictive variables [35,36]. Building on a classical KNN approach, we constructed a model able to predict clinical evolution at 3 months with 84% accuracy. While validation with larger and external cohorts remains necessary, such tools may be valuable for informing patients and their relatives about their mid-term clinical outcomes.

Our study underscores the significant role of IONM in predicting as well as in mitigating the risk of postoperative deficit. Critics have questioned the level of evidence regarding the utility of monitoring due to the lack of prospective studies with non-monitored control groups [37]; however, ethical considerations preclude such study designs. Despite this limitation, consensus among experts supports the utility of monitoring in spinal cord and spine surgery, as evidenced by retrospective studies [38]. While the role of IONM in predicting postoperative deficit is widely accepted, its effectiveness in limiting this deficit is more challenging to demonstrate [19,20,29]. While our study does not provide a direct answer to this question, it presents arguments in favor of a preventive role for alerts; when the surgeon receives real-time warnings of minor alterations in evoked potentials during the procedure, various measures can be taken (such as correction of arterial hypotension, warming of the surgical site with warm saline, or moving the surgical site to another area). If the alert is major (such as abolition of MEPs and/or reduction of the D wave by more than 50%) and there is no recovery within minutes, then the procedure is interrupted [10]. In our study, 26% (10/38) of alerts were only transient; following the described measures, none of these alerts resulted in postoperative worsening. This suggests a protective role for monitoring, with an alert prompting the surgeon to take corrective measures to reduce postoperative clinical worsening. However, the precise time window during which the at-risk tissue can be salvaged remains unknown.

Lastly, criticisms of intraoperative monitoring have been raised regarding concerns that excessive alerts may limit tumor resection. However, these arguments have been refuted in a recent review, which emphasized that IONM alerts do not curtail the extent of resection [39]. In our study, 79% of patients underwent total tumor resection. Furthermore, no correlation was found between the occurrence of definitive alerts and the presence of tumor residue. Therefore, the inability to achieve total tumor resection was more closely related to direct surgical constraints rather than IONM alerts.

This study is subject to certain limitations that merit consideration. Medium-term clinical outcome following spinal tumor surgeries is intricately linked to the proficiency of the surgical team, and consequently to the nature of the procedure undertaken. However, a comprehensive evaluation of the surgical intervention itself proved challenging due to the variability in the employed surgical approaches, which defy simple quantitative characterization. Moreover, the nuanced prognostic impact of specific MRI lesion characteristics could not be exhaustively assessed beyond the rostro-caudal dimensions of the lesion, though such aspects warrant a dedicated and focused investigation in the future.

## 5. Conclusions

Despite advancements in surgical and imaging techniques, spinal cord tumors (SCTs) continue to pose a significant surgical challenge. Over the years, IONM has emerged as a powerful method for real-time assessment of motor and somatosensory spinal tracts during tumor resection. The occurrence of an alert during IONM is strongly correlated with increased risk of neurological deterioration, particularly if the alert persists throughout surgery. Therefore, IONM should be routinely performed to promptly detect spinal cord lesions that may be reversible and preserve the functionality of the spinal tracts. The detection of spinal lesions does not result in excessive or insignificant alerts, as tumor resection is successfully carried out in the majority of patients. Lastly, surgical outcomes can be estimated using multivariate and machine learning models which incorporate clinical, radiological, histological, and electrophysiological data; however, it is essential to validate these models with larger surgical cohorts to ensure their reliability and accuracy. 

## Figures and Tables

**Figure 1 cancers-16-02192-f001:**
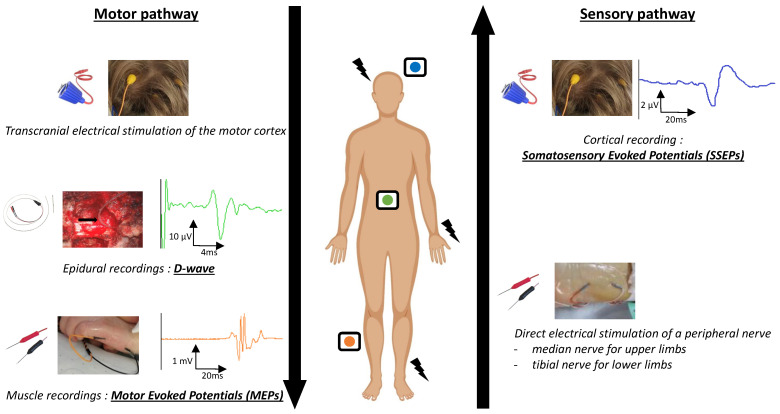
Schematic view of the stimulation and recording points during intraoperative neuromonitoring for spinal cord tumors. Motor evoked potentials (MEPs): Transcranial electrical stimulation of the primary motor cortex (using trains of stimulations delivered with corkscrew electrodes at C1 and C2 according to the international 10–20 EEG system) with muscular recordings (numbers and location of the muscles pairs depend on tumor location). D-Wave: Transcranial electrical stimulation of the primary motor cortex with epidural recording caudal to the operation site. Somatosensory evoked potentials (SSEPs): Direct electrical stimulation of a peripheral nerve and cortical recordings of the primary somatosensory cortex (recorded at P3/P4 or C’z for upper and lower limbs, respectively).

**Figure 2 cancers-16-02192-f002:**
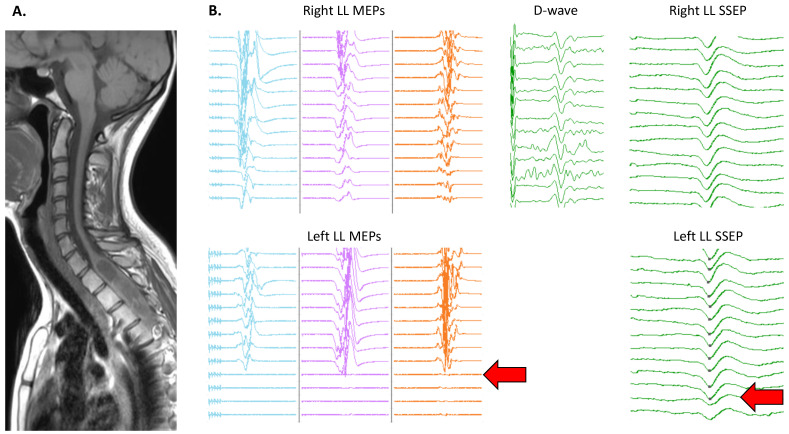
Example of intraoperative neuromonitoring (IONM): (**A**) pre-operative sagittal T1-weighted MRI scans showing an intradural intramedullary dorsal tumor; (**B**) IONM screenshots with motor evoked potentials (MEPs) recording in vastus lateralis, tibialis anterior, and extensor digitorum brevis muscles, along with D-wave recordings and somatosensory evoked potentials (SSEPs) after tibial nerve stimulation. During tumor resection, two alert criteria were identified for the left lower limb (indicated by arrows), identified by a sudden loss of MEPs across all recorded muscles (without any modification in D-waves) followed by a 50% reduction in SSEP amplitude. Both alerts were transient, and the patient did not experience postoperative deficits.

**Table 1 cancers-16-02192-t001:** Patients’ characteristics and overall neurophysiological results.

	All Patients (*n* = 67)
**Age (Years)**	50 (14)
**Gender**	
Male	43 (64%)
Female	24 (36%)
**Lesion size (mm)**	32 (20)
**Preoperative McCormick Scale**	
Mild	44 (64%)
Severe	24 (36%)
**Tumor level**	
High	32 (48%)
Low	35 (52%)
**Tumor location**	
Intradural extramedullary	40 (60%)
Intradural intramedullary	27 (40%)
**Preop Evoked Potentials**	
Normal	42 (63%)
Altered	25 (37%)
**IONM any alert**	
No Alert	29 (43%)
Alert	38 (57%)
**IONM persistent alert**	
No alert	39 (58%)
Alert	28 (42%)
**Histology**	
Ependymoma	24 (35%)
Schwannoma	14 (21%)
Meningioma	17 (25%)
Other	12 (18%)
**MRI residual lesion**	
No	52 (79%)
Yes	14 (21%)

Categorical variables are expressed as *n* (%) and continuous variables as means (standard deviation).

**Table 2 cancers-16-02192-t002:** Prognostic value of IONM alerts at the patient level.

IONM Alerts	Sensitivity	Specificity	PPV	NPV	Accuracy
transient alert MEPs-DW	0	0.85	0	0.91	0.79
transient alert SSEPs	0	0.90	0	0.78	0.73
transient alert MEPs-DW & SSEPs	0	0.72	0	0.96	0.70
any alert MEPs-DW	0.86	0.76	0.63	0.92	0.79
any alert SSEPs	0.6	0.75	0.54	0.79	0.70
persistent alert MEPs-DW	0.86	0.89	0.79	0.93	0.88
persistent alert SSEPs	0.55	0.92	0.78	0.80	0.80
any alert MEPs-DW & SSEPs	0.95	0.62	0.55	0.96	0.73
persistent alert MEPs-DW & SSEPs	0.95	0.84	0.75	0.97	0.88

Data are expressed as %.

**Table 3 cancers-16-02192-t003:** Predictors of neurological deterioration at the patient level.

Predictors	Good Outcome (*n* = 45)	Bad Outcome (*n* = 22)	Univariate (*p*-Value)	Multivariate (*p*-Value)
**Age (years)**	50 (15)	49 (12)	NS	
**Gender**				
Male	15 (33%)	9 (41%)	NS	
Female	30 (67%)	13 (59%)		
**Lesion size (mm)**	29 (19)	39 (19)	0.01	NS
**Preop McCormick Scale**				
Mild	30 (67%)	14 (64%)	NS	
Severe	15 (33%)	8 (36%)		
**Tumor level**				
High	17 (38%)	15 (68%)	0.03	NS
Low	28 (62%)	7 (32%)		
**Tumor location**				
Intradural extramedullary	30 (67%)	10 (45%)	NS	
Intradural intramedullary	15 (33%)	12 (55%)		
**Preop evoked potentials**				
Normal	33 (73%)	9 (41%)	0.02	0.04
Altered	12 (27%)	13 (59%)		
**IONM any alert**				
No alert	28 (62%)	1 (4%)	<0.001	<0.001
Alert	17 (38%)	21 (95%)		
**IONM persistent alert**				
No alert	38 (84%)	1 (5%)	<0.001	<0.001
Alert	7 (16%)	21 (95%)		
**Histology**				
Ependymoma	10 (22%)	14 (64%)		
Schwannoma	11 (24%)	3 (14%)	0.001	NS
Meningioma	14 (31%)	3 (14%)		
Other	10 (22%)	2 (9%)		
**MRI Residual lesion**				
No	37 (82%)	15 (71%)	NS	
Yes	8 (18%)	6 (29%)		

Categorical variables are expressed as *n* (%) and continuous variables as means (standard deviation). *p*-value is provided for statistically significant results; NS, non-significant.

**Table 4 cancers-16-02192-t004:** Prognostic value of IONM alerts at the individual limb level.

IONM Alerts	Sensitivity	Specificity	PPV	NPV	Accuracy
any alert SSEPs	0.42	0.83	0.42	0.83	0.73
any alert MEPs-DW	0.78	0.82	0.58	0.92	0.81
any alert MEPs-DW & SSEPs	0.81	0.72	0.48	0.92	0.74

Data are expressed as %.

**Table 5 cancers-16-02192-t005:** Predictors of neurological deterioration at the individual limb level.

Predictors	Stable (*n* = 152)	Worsening (*n* = 48)	Univariate (*p*-Value)	Multivariate (*p*-Value)
**Lesion size (mm)**	32 (21)	40 (18)	<0.001	0.02
**Tumor level**				
High	88 (58%)	38 (79%)	0.01	NS
Low	64 (42%)	10 (21%)		
**Preop evoked potentials**			NS	NS
Normal	55 (36%)	23 (48%)		
Altered	97 (64%)	25 (52%)		
**IONM alert**				
No alert	100 (72%)	9 (19%)	<0.001	<0.001
Alert	42 (28%)	39 (81%)		
**Histology**				
Ependymoma	50 (33%)	34 (71%)		
Schwannoma	30 (20%)	4 (8.3%)	<0.001	NS
Meningioma	43 (28%)	5 (10%)		
Other	29 (19%)	5 (10%)		
**MRI Residual lesion**				
No	127 (84%)	31 (70%)	NS	NS
Yes	25 (16%)	13 (30%)		

Categorical variables are expressed as *n* (%) and continuous variables as means (standard deviation). *p*-value is provided for statistically significant results; NS, non-significant.

## Data Availability

The data presented in this study are available on request from the corresponding author.

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
