# Peer review of "The Impact of Neurophysiological Monitoring during Intradural Spinal Tumor Surgery"

_cancers, 2024, doi:10.3390/cancers16122192_

Round 1

Reviewer 1 Report

Comments and Suggestions for Authors

Good study

Why do preoperative monitoring? Isn't this uncomfortable to the patient? Why not just baseline obtained after anesthesia and before cutting skin?

Discuss the value of wake up test during surgery if persistent changes in IONM occur.

IONM includes the word monitoring, so no need to say IONM monitoring.

Neurinoma is more French, better use schwannoma or neurofibroma (WHO terminology).

34/84 ependymomas had worsening of neurological function, that number is too high for a tumor with good margins. Usually astrocytomas have worse prognosis since have no good margins. How do the authors explain this discrepancy between their findings and the general literature?

The conclusion that spinal lesions did not hamper oncological prognosis is very vague and not supported by data from this paper. Is this about spine or spinal cord? Primary tumors or metastases?

Comments on the Quality of English Language

minor edits

Author Response

Dear reviewer,

We would like to thank you for the exhaustive review of our paper. Please find here a point-by-point reply to your questions.

“Why do preoperative monitoring? Isn't this uncomfortable to the patient? Why not just baseline obtained after anesthesia and before cutting skin? “

We agree with the reviewer that preoperative neurophysiological monitoring are not commonly performed in centers performing surgery for spinal cord tumors.

Indeed, in many centers, evoked potentials are only recorded at the beginning of surgery, after incision.

Preoperative monitoring is not uncomfortable since MEPs are performed with transcranial magnetic intensity (and not electrical stimulation at high intensity such as the stimulation procedure used during IONM) and SSEPs are performed with median or tibial nerve stimulation at low intensity.

However, in our opinion, there are several reasons to perform preoperative evoked potentials before surgery : i) preoperative evoked potentials may be used to detect neurological dysfunction preoperatively, a factor that may guide surgical indication in slowly evolving tumors. ii) secondly, as shown by multivariate analysis alterations of preoperative evoked potentials are associated with an increased risk of post-operative clinical deterioration. We believe that this information may be useful to inform both patients and families before surgery and the neurosurgeon of the neurological risk of the patient. Iii) Preoperative evoked potentials are helpful to assess the feasibility of IONM. Indeed, in our experience, major alterations of preoperative MEPs of SSEPs preclude the possibility to obtain reproductible responses during surgery. In the present study, all patients had at least either cortical SSEPs or MEPs before surgery and IONM was feasible for all patients iv) Lastly, when the patient has a post-operative functional complaint that does not always correlate with the clinical examination, the surgeon requests evoked potentials, but it is impossible to assess the responsibility of surgery in the abnormalities observed in the absence of a pre-operative reference. We have added a sentence (written in red font) in the discussion to address this point.

“Discuss the value of wake up test during surgery if persistent changes in IONM occur.”

Wake-up test is commonly performed during scoliosis surgery. Major complication of scoliosis surgery includes paraplegia or severe motor disorders and wake-up test is still performed for spinal deformation surgeries. However, for spinal cord tumors surgery, lesions of the motor tracts is often more limited and progressive or concerns only one limb and the wake-up test may be falsely negative. Moreover, surgery of spinal tumors often relies on microsurgery, and the wake-up might have a negative impact on the surgical procedure itself. For spinal cord tumors surgery, wake-up test is nowadays more often replaced by IONM which detects more easily minor spinal cord lesions. However, the wake-up test may be performed when the IONM results are unreliable and when the IONM results show changes but questions remain about the patient’s motor status.

We have added a sentence in the discussion (written in red font) to address this point.

“IONM includes the word monitoring, so no need to say IONM monitoring.”

We agree with the reviewer’s suggestion. We changed the manuscript accordingly.

“Neurinoma is more French, better use schwannoma or neurofibroma (WHO terminology).”

We agree with the reviewer’s suggestion. We changed the manuscript accordingly.

“34/84 ependymomas had worsening of neurological function, that number is too high for a tumor with good margins. Usually astrocytomas have worse prognosis since have no good margins. How do the authors explain this discrepancy between their findings and the general literature?”

In our cohort, we included only patients with pure intramedullary ependymoma. We did not include patients with ependymoma with an extension of the lesion in the extramedullary space. According to the literature, those patients are at high risk of neurological dysfunction and indeed, 14/24 ependymoma patients of our cohort had a neurological deterioration. However, it should be noticed that the criteria to detect neurological deterioration were highly sensitive : any increase of the MMS (even for 1 point) or somatosensory deficit was considered as a clinical deterioration. Using more strict criteria (such as severe deterioration with difficulty in walking ability) would decrease the rate of neurological deterioration and lead to results closer to other case series. Moreover, the clinical status was evaluated 3 months post-operatively and functional recovery can still occur after that period. We  changed the manuscript accordingly (written in red font in the manuscript).

“The conclusion that spinal lesions did not hamper oncological prognosis is very vague and not supported by data from this paper. Is this about spine or spinal cord? Primary tumors or metastases?”

We only included patients with intradural tumors including mostly ependymomas, meningiomas and schwannomas (see table I). Only two patients had a pilocytic astrocytomas and 1 patient metastasis of a lung carcinoid tumor (see section 3.1 in the results).

We agree with the reviewer’s suggestion that the sentence “spinal lesions did not hamper oncological prognosis is vague” and changed it accordingly (written in red font in the discussion)

Reviewer 2 Report

Comments and Suggestions for Authors

The authors present a study demonstrating the pivotal role of pre- and intraoperative neurophysiological explorations in detecting and preventing neurological suffering during spinal tumour surgery.

It is a retrospective study, although they do not say so explicitly (they should add it in the material and methods), in which approval was obtained from the ethics committee last December 2023. In these two and a half months they have analysed the data and prepared the manuscript. Excellent work.

The statistical design should clarify that it analyses paired data, preoperative and intraoperative in each patient, to look for an association with postoperative results at 3 months. The multivariate logistic regression model is dangerous to use, because with only 67 patients it has little statistical power.

The Tables and Figures are adequate. In Table 1 I would add capital letters at the beginning of each word. The first verifiable (age (years) 50 (14) ) is not understood. They should clarify whether it is mean and standard deviation, which is what it looks like, or median and interquartile range. These same considerations should be taken into account in Table 3 and 5. Variables where the possibilities are YES/NO (residual lesion, for examplei), can be simplified to make the tables less dense.

Otherwise, I recommend a thorough review of the manuscript by a native speaker to check for grammatical and verb tense errors in the manuscript.

Comments on the Quality of English Language

I recommend a thorough review of the manuscript by a native speaker to check for grammatical and verb tense errors in the manuscript.

Author Response

Dear reviewer,

We would like to thank you for the exhaustive review of our paper. Please find here a point-by-point reply to your questions.

“The statistical design should clarify that it analyses paired data, preoperative and intraoperative in each patient, to look for an association with postoperative results at 3 months.”

We agree with the reviewer that preoperative evoked potentials and peroperative evoked potentials are related to the same patient. However, evoked potentials recorded preoperatively and during IONM are not recorded with the same procedure (with transcranial magnetic stimulation for preoperative MEPs and with electrical cortical stimulation during IONM : in other words, they do not represent the same measure. Moreover, they may be uncorrelated since alerts during IONM may represent specific surgical events not related to preoperative data. For those reasons, we believe that including those two variables in the logistic regression model brings significant informative information.

As an exploratory analysis however, to account for a possible link between preoperative evoked potentials and peroperative evoked potentials for multivariate analysis we also tested a Generalized Linear Mixed Model (GLMM) using lme4 R package (https://search.r-project.org/CRAN/refmans/lme4/html/glmer.html). GLMMs are an extension of Generalized Linear Models that incorporate random effects to account for correlation and hierarchical structure within the data. Subject identity can be included as a random effect in GLMMs when dealing with repeated measures or clustered data, where observations are not independent due to grouping of data points within subjects. For this analysis, fixed effects variables were evoked potentials, size of the tumor, tumor level, histology and presence of MRI residual lesion while subject identity was the random effect variable.

The same significant variables as those obtained with a classical GLM were found but were not added within manuscript to keep clarity for the reader.

“The multivariate logistic regression model is dangerous to use, because with only 67 patients it has little statistical power.”

We agree with the reviewer that population size must be taken into account in the choice of statistical model, and we have applied the one in ten rule. In statistics, the one in ten rule is a rule of thumb for how many predictor parameters can be estimated from data when doing regression analysis (in particular proportional hazards models in survival analysis and logistic regression) while keeping the risk of overfitting and finding spurious correlations low. The rule states that one predictive variable can be studied for every ten events (Peduzzi P, Concato J, Kemper E, Holford TR, Feinstein AR. A simulation study of the number of events per variable in logistic regression analysis. J Clin Epidemiol. 1996 Dec;49(12):1373-9. doi: 10.1016/s0895-4356(96)00236-3. PMID: 8970487). Since 67 observations were included in the analysis with 6 predicting we consider that our model has reasonable power to evaluate the relation between clinical outcome at 3 months and predicting variables.

Moreover, we found that several variables had high coefficient estimates with significant p-value suggesting that the model had enough explaining value.

“The Tables and Figures are adequate. In Table 1 I would add capital letters at the beginning of each word. The first verifiable (age (years) 50 (14) ) is not understood. They should clarify whether it is mean and standard deviation, which is what it looks like, or median and interquartile range. These same considerations should be taken into account in Table 3 and 5. Variables where the possibilities are YES/NO (residual lesion, for example), can be simplified to make the tables less dense.”

We agree with the reviewer’s suggestion to add capital letters at the beginning of each variable. In table 1, 3 and 5, a footnote now clarifies that categorical data are expressed as n (\%) and continuous variables as means (standard deviation.

“Otherwise, I recommend a thorough review of the manuscript by a native speaker to check for grammatical and verb tense errors in the manuscript.”

We revised the manuscript with a native speaker to check for grammatical and verbe tense errors.

Reviewer 3 Report

Comments and Suggestions for Authors

Although the title is spinal cord tumor, most are extramedullary tumors. MRI findings and surgery itself must be included in this analysis.

Neurophysiological monitoring are well known method to evaluate spinal cord functions.  What is the new findings of this study?

Author Response

Dear Reviewer,

We would like to thank you for the exhaustive review of our paper. Please find here a point-by-point reply to your questions.

“Although the title is spinal cord tumor, most are extramedullary tumors.”

In the present study, we included only patients with intradural tumors since they are associated with high functional risk following surgery. We agree with the reviewer that spinal cord tumors may be misleading since not all tumors were intramedullary tumors. Accordingly, we changed the title to “the impact of pre and intraoperative monitoring of motor and somatosensory pathways during surgery of intradural spinal tumors”.

 “MRI findings and surgery itself must be included in this analysis.”

We agree with the reviewer that both MRI findings and surgery itself should also be included in the analysis. Accordingly, the statistical analysis that we conducted included both factors with available data. MRI findings were indeed included in univariate and multivariate analysis, taking into account the rostro-caudal size of the lesion. Moreover, presence of MRI residual lesion, a factor directly related to the extent of surgical resection, was included in the analysis. 

“Neurophysiological monitoring are well known method to evaluate spinal cord functions.  What is the new findings of this study?”

We agree with the reviewer that several studies have shown the relevance of neurophysiological monitoring to evaluate spinal cord functions. However, we believe that our study adds several new insights to available evidence : i) in most studies, the prognostic value of IONM alterations is described and evaluated in isolation. The prognostic of spinal tumors is complex and depends on several factors including clinical variables, MRI findings, histology, neurophysiological data. This was precisely the objective of multivariate analysis and of the machine learning model  of the data. We show here that IONM alterations have prognostic significance independently of other factors, a finding that highlights their specific value. Ii) we show that IONM alterations predict functional status both at the patient level and at individual limb level, highlighting their anatomical specificity. Iii) lastly, we show that preoperative evoked potentials are also relevant to predict clinical outcome and to anticipate alterations during surgery.

Round 2

Reviewer 2 Report

Comments and Suggestions for Authors

The authors have made the necessary corrections, following the recommendations of the reviewers.

The quality of the tables and images in the manuscript is adequate.

The references in the manuscript are also adequate for this study.

From my point of view, the manuscript is ready for publication in the current version.

Author Response

Dear Reviewer,

Thank you again for your review with careful suggestions. I hope the study is now ready for publications

Kind regards

J Jung

Reviewer 3 Report

Comments and Suggestions for Authors

Still, the title is intramedullary spinal tumors.

Study limitations should be discussed.

Comments on the Quality of English Language

NA

Author Response

Dear reviewer,

 Please find here a point-by-point reply to your questions.

“Still, the title is intramedullary spinal tumors. ”

We acknowledge there was a mistake in the change made to the title. We changed it to the impact of pre and intraoperative monitoring of motor and somatosensory pathways during surgery of intradural spinal tumors”.

“Study limitations should be discussed..”

We added a paragraph to the discussion section to address certain limitations of the study (highlighted in red in the text). In particular, we mentioned that the prognostic influence of the surgical procedure itself could not be fully evaluated because the surgical approaches employed varied and were difficult to describe by a simple quantitative parameter. We also noted that detailed characteristics of the lesions on MRI could not be assessed, aside from the rostro-caudal size of the lesion, but could be the subject of a dedicated separate study.